# Iodine Status Modifies the Association between Fluoride Exposure in Pregnancy and Preschool Boys’ Intelligence

**DOI:** 10.3390/nu14142920

**Published:** 2022-07-16

**Authors:** Carly V. Goodman, Meaghan Hall, Rivka Green, Jonathan Chevrier, Pierre Ayotte, Esperanza Angeles Martinez-Mier, Taylor McGuckin, John Krzeczkowski, David Flora, Richard Hornung, Bruce Lanphear, Christine Till

**Affiliations:** 1Department of Psychology, York University, Toronto, ONT M3J 1P3, Canada; cgoodman@yorku.ca (C.V.G.); mkhall@yorku.ca (M.H.); rrgreen@yorku.ca (R.G.); mcguckin@yorku.ca (T.M.); krzeczkj@yorku.ca (J.K.); dflora@yorku.ca (D.F.); 2Department of Epidemiology, Biostatistics, and Occupational Health, Faculty of Medicine and Health Sciences, McGill University, Montreal, QC H3A 1G1, Canada; jonathan.chevrier@mcgill.ca; 3Département de Médecine Sociale et Préventive, Faculté de Médecine, Université Laval, Québec, QC G1V 0A6, Canada; pierre.ayotte@inspq.qc.ca; 4Cariology, Operative Dentistry and Dental Public Health, Indiana University School of Dentistry, Indianapolis, IN 46202, USA; esmartin@iupui.edu; 5Pediatrics and Environmental Health, Cincinnati Children’s Hospital Medical Center, Cincinnati, OH 45229, USA; rwhornung@yahoo.com; 6Faculty of Health Sciences, Simon Fraser University, Burnaby, BC V5A 1S6, Canada; bruce_lanphear@sfu.ca

**Keywords:** fluoride, iodine, intelligence, neurodevelopment, pregnancy

## Abstract

In animal studies, the combination of in utero fluoride exposure and low iodine has greater negative effects on offspring learning and memory than either alone, but this has not been studied in children. We evaluated whether the maternal urinary iodine concentration (MUIC) modifies the association between maternal urinary fluoride (MUF) and boys’ and girls’ intelligence. We used data from 366 mother–child dyads in the Maternal–Infant Research on Environmental Chemicals Study. We corrected trimester-specific MUF and MUIC for creatinine, and averaged them to yield our exposure variables (MUF_CRE_, mg/g; MUIC_CRE_, µg/g). We assessed children’s full-scale intelligence (FSIQ) at 3 to 4 years. Using multiple linear regression, we estimated a three-way interaction between MUF_CRE_, MUIC_CRE_, and child sex on FSIQ, controlling for covariates. The MUIC_CRE_ by MUF_CRE_ interaction was significant for boys (*p* = 0.042), but not girls (*p* = 0.190). For boys whose mothers had low iodine, a 0.5 mg/g increase in MUF_CRE_ was associated with a 4.65-point lower FSIQ score (95% CI: −7.67, −1.62). For boys whose mothers had adequate iodine, a 0.5 mg/g increase in MUF_CRE_ was associated with a 2.95-point lower FSIQ score (95% CI: −4.77, −1.13). These results suggest adequate iodine intake during pregnancy may minimize fluoride’s neurotoxicity in boys.

## 1. Introduction

Fluoride exposure during early brain development has been associated with diminished intelligence quotient (IQ) scores among children living in areas with high levels of naturally occurring fluoride in drinking water (~3 mg/L) [1,2,3] and in areas where fluoride is added to public water supplies or salt for caries prevention [4,5,6]. The mechanism(s) underlying fluoride-associated cognitive deficits are not well understood, but changes to the thyroid function may be one such mechanism [7,8,9,10,11]. In 2006, the National Research Council (NRC) classified fluoride as an endocrine disruptor and recommended more research to understand fluoride’s effects on the thyroid gland, especially in iodine deficient pregnant women [12].

Iodine is an essential nutrient for thyroid hormone synthesis and normal thyroid function [13]. Sufficient iodine intake is critical for optimal maternal and fetal thyroid function and normal fetal neurodevelopment [14,15,16,17]. Even mild to moderate iodine deficiency in pregnancy has been linked to diminished cognitive abilities in children [14,18,19,20,21,22,23,24], though not in all studies [25,26]. The inconsistent results may reflect differences in the severity of maternal iodine deficiency, methodology, age at outcome assessment, or other biological co-factors.

Studies conducted in China examined whether fluoride exposure and iodine deficiency combine to impart adverse effects on children’s intelligence. Notably, school-aged children living in endemic fluoride and iodine-deficient areas had lower IQ scores than those living in endemic fluoride areas alone or iodine-deficient areas alone [27,28]. Fluoride in drinking water was reported to exacerbate the adverse effects of low iodine on child neurodevelopment and central nervous system function more broadly [28]. However, these studies were cross-sectional and did not account for potential confounders. In experimental studies, rat offspring exposed to both high fluoride and low iodine in utero showed greater deficits in learning and memory compared with those exposed to either high fluoride or low iodine [29,30].

Given the ubiquity of fluoride exposure, along with recent trends showing mild-to-moderate iodine deficiency in pregnant women [17,31,32], we evaluated whether the maternal iodine status modifies the association between prenatal fluoride exposure and children’s intelligence. We hypothesized that low urinary iodine concentrations in Canadian pregnant women would exacerbate the fluoride-associated intellectual deficits observed in their children. We further hypothesized that the effects would be stronger in boys than girls given previous findings of sex differences in the neurotoxicity of prenatal fluoride exposure [33].

## 2. Materials and Methods

### 2.1. Participants

Participants included mother–child dyads enrolled in the Canadian Maternal–Infant Research on Environmental Chemicals (MIREC) study. Between 2008 and 2011, 2001 pregnant women were recruited from 10 cities across Canada to participate in a longitudinal cohort study. The inclusion criteria were as follows: women who were 18 years of age or older who could provide consent, communicate in English or French, and were <14 weeks’ gestation. Participants were excluded if they had any medical complications, any known fetal abnormalities, or if there was illicit drug or alcohol abuse during pregnancy. Additional details of the MIREC study can be found in the cohort profile [34].

A subset of 808 women provided consent to participate with their child in the MIREC-Child Development Plus (CD Plus) follow-up study. Due to budgetary constraints, recruitment for MIREC CD Plus was limited to six of the ten cities from the original cohort, namely Vancouver, Toronto, Hamilton, Montreal, Kingston, and Halifax. The inclusion criteria for mother–child dyads in MIREC-CD Plus were as follows: mothers of singleton children born >28 weeks’ gestation who were between the ages of 3 and 4 at time of the study and had no congenital abnormalities, major neurological disorders, or history of convulsions. Among the 808 women who consented, 610 agreed to child IQ testing (76%), 601 of whom completed the neurodevelopmental testing. The latter subset of 601 mother–child dyads provided data for the current study.

Of the 601 children who completed IQ testing, 366 had complete data on maternal urinary fluoride (MUF), maternal urinary iodine concentration (MUIC), urinary creatinine (CRE), and covariates (See Figure 1); 235 were excluded for missing (i) creatinine data at all three trimesters (*n* = 175), (ii) a valid MUF measure available at all three trimesters (*n* = 9), (iii) MUIC_CRE_ < 600 μg/g data at trimesters 1 and 2 (*n* = 40), and (4) covariate data (*n* = 11). Women with MUIC_CRE_ values greater than or equal to 600 μg/g (*n*= 37) were excluded from the analyses because excess iodine levels have been linked to diminished intelligence [35], and we were specifically interested in comparing women with “low” levels of iodine with those with “adequate” levels of iodine, rather than “excess” levels of iodine. We considered MUIC_CRE_ values greater than or equal to 600 μg/g to be “higher than adequate” as opposed to the WHO cut-off of 500 μg/L for unadjusted MUIC [36], given that we used MUIC values corrected for creatinine, and MUIC_CRE_ values increase from trimester 1 to 2 [37].

The present study was approved by the research ethics boards at Health Canada and York University. The MIREC study was also approved by the research ethics boards at all participating recruitment sites and at Health Canada. All participants provided their informed consent.

### 2.2. Urine Collection

Urine was collected in Nalgene^®^ containers, labeled with a unique identification, aliquoted into smaller Cryovials^®^, and stored at appropriate temperatures until they were shipped for fluoride or iodine analysis. Spot samples were collected in each trimester of pregnancy at a mean ± SD of 11.57 ± 1.57, 19.11 ± 2.39, and 33.11 ± 1.50 weeks’ gestation.

### 2.3. Maternal Urinary Fluoride Concentration

We derived maternal urinary fluoride (MUF, mg/L) concentrations by averaging fluoride concentrations across trimesters. We previously found a moderate correlation between the three samples, with intraclass correlation coefficients ranging from 0.37 to 0.40 [38]. Urine samples were analyzed at the Indiana University School of Dentistry through a modification of the hexamethyldisiloxane (HMDS; Sigma Chemical Co., St. Louis, MO, USA) micro-diffusion procedure described previously [38,39]. In neutral solutions, fluoride concentrations were measured down to 0.02 mg/L. Two of the spot urine samples (0.04%) were excluded from the first trimester as the readings surpassed the highest concentration standard of the instrument (5 mg/L).

### 2.4. Maternal Urinary Iodine Concentration

We derived the maternal urinary iodine concentration (MUIC, µg/L) by averaging iodine concentrations from two spot urine samples collected in the first and second trimester. MUIC is considered a reliable biomarker of recent iodine intake and reflects total iodine intake from all dietary sources [40]. MUIC was measured by the accredited Toxicology Laboratory at the Institut National de Santé Publique du Québec (INSPQ) using inductively coupled plasma mass spectrometry (ICP-MS). Values below the limit of detection (LOD, 38 µg/L) were replaced with the LOD divided by the square root of 2 (Hornung and Reed, 1990); 180 (9.74%) and 79 (4.56%) mothers had a MUIC below the LOD in trimesters 1 and 2, respectively.

### 2.5. Correcting for Variability in Urinary Dilution

To account for variability in urine dilution at time of measurement, MUF and MUIC were corrected for creatinine (CRE) measured in the same spot sample using the following equations:$${\mathrm{MUF}}_{\mathrm{CRE}\;(\mathrm{mg}/g)}=\frac{(MUF_{T1}/CRE_{T1})+\left({MUF}_{T2}/CRE_{T2}\right)+\left(MUF_{T3}/CRE_{T3}\right)}{3},$$
$${\mathrm{MUIC}}_{\mathrm{CRE}\;(\mu g/g)}=\frac{\left(MUIC_{T1}/CRE_{T1}\right)+\left(MUIC_{T2}/CRE_{T2}\right)}{2},$$
where MUF_T1_ is the observed fluoride concentration, MUIC_T1_ is the observed MUIC, and CRE_T1_ is the observed creatinine concentration for that individual in trimester 1. MUF_T2_ is the observed fluoride concentration, MUIC_T2_ is the observed MUIC, and CRE_T2_ is the observed creatinine concentration for that individual in trimester 2. MUF_T3_ is the observed fluoride concentration, and CRE_T3_ is the observed creatinine concentration for that individual in trimester 3. The measurement of urinary creatinine was previously described [38]. In pregnant women, the iodine to creatinine ratio (MUIC/CRE) is moderately correlated with 24 h urinary iodine excretion, the gold standard measure of iodine status relative to uncorrected urinary iodine concentration [14,41].

### 2.6. Children’s Full-Scale Intelligence Quotient

We assessed children’s intellectual abilities at 3 to 4 years of age using the Wechsler Preschool and Primary Scale of Intelligence-III with Canadian age-standardized norms (mean = 100, SD = 15). The test was administered in children’s homes in either English or French by qualified research professionals who were blinded to gestational iodine status or fluoride exposure in pregnancy. We used full-scale intelligence (FSIQ), a measure of global intellectual and cognitive functioning, as our primary outcome.

### 2.7. Covariates

We selected covariates a priori based on prior work with the same study cohort examining fluoride exposure and children’s intellectual abilities [6]. Covariates included maternal education (dichotomized as bachelor’s degree or higher), maternal race (White/non-White), study site, and a continuous measure of the quality of the home environment using the Home Observation for Measurement of the Environment (HOME)—Revised Edition [42] at the time of the home visit when children were aged 3 to 4 years old.

### 2.8. Statistical Analyses

We used chi-square tests for categorical covariates and t-tests for continuous covariates to test for sampling differences between those with complete data and those without complete data (i.e., without MUF_CRE_ or MUIC_CRE_ but with FSIQ data). For descriptive purposes, MUIC_CRE_ was stratified into those with low (<200 μg/g) and adequate ≥200 and <600 μg/g urinary iodine. Independent sample t-tests were used to test for differences between boys and girls for exposure and outcome variables. Welch’s correction was applied for t-tests to account for unequal variance.

We used multiple linear regression to estimate a model with a three-way interaction between MUF_CRE_, MUIC_CRE_, and child sex in predicting children’s FSIQ scores while controlling for maternal education, maternal race, study site, and the HOME score; this model included all constituent two-way interaction terms and first-order effects. To determine whether MUF_CRE_ and MUIC_CRE_ interact as a function of sex without stratifying the sample, we then examined the model-implied MUF_CRE_ by MUIC_CRE_ two-way interaction within each sex. To facilitate the interpretation of coefficients, we centered MUIC_CRE_ (i.e., subtracted a constant from every value of MUIC_CRE_) around a “low” level (i.e., 147 μg/g which corresponds to the 10th percentile value for MUIC_CRE_) and an “adequate” level (i.e., 294 μg/g which corresponds to the 50th percentile value for MUIC_CRE_) [43,44]. We then re-ran the model using MUIC_CRE_ centered around the “low” and “adequate” levels of iodine, separately, with boys coded as the reference. The model was re-estimated with girls coded as the reference to interpret the association between MUF_CRE_ and FSIQ for a girl whose mother had a low or adequate level of iodine during pregnancy. All models were evaluated for linearity, homoscedasticity, and normality and model assumptions were sufficiently met. No influential outliers were detected according to Cook’s distance.

We used STATA version 16.1 (STATA corporation) for data analysis. The level of significance was 0.05, and all statistical tests were two-tailed. All coefficients were reported for every 0.5 mg/g increase in MUF_CRE_ (approximately the IQR).

## 3. Results

Most participants included in the present study were married or in a common-law relationship, had a bachelor’s degree or higher, and were White (Table 1). Mother–child dyads with complete data did not significantly differ from those without complete data on any of the demographic characteristics, except a greater proportion of mothers with complete data were White.

The median (IQR) MUF_CRE_ and MUIC_CRE_ were 0.61 (0.49) mg/g and 294 (181) μg/g, respectively. Boys and girls did not differ significantly in MUF_CRE_ concentration or MUIC_CRE_ (Table 2). Children’s FSIQ scores were in the average range, with girls scoring significantly higher than boys (*t*(364) = −3.17, *p* = 0.002; Table 2).

### Three-Way Interaction Model

We found a significant three-way interaction between MUF_CRE_, MUIC_CRE_, and sex while controlling for relevant covariates (*p*  =  0.019; see Table 3 and Figure 2). The two-way MUIC_CRE_ by MUF_CRE_ interaction was significant for boys (*p* = 0.042), but not girls (*p* = 0.190). For boys whose mothers had a low MUIC_CRE_, every 0.5 mg/g increase in MUF_CRE_ was associated with a 4.65-point lower FSIQ score (95% CI: −7.67, −1.62; *p* = 0.003). For boys whose mothers had adequate MUIC_CRE_, every 0.5 mg/g increase in MUF_CRE_ was associated with a 2.95-point lower FSIQ score (95% CI: −4.77, −1.13; *p* = 0.002). In contrast, MUF_CRE_ was marginally associated with FSIQ for girls whose mothers had low MUIC_CRE_ (B = 2.48; 95% CI: −0.31, 5.26; *p* = 0.081) and was not significantly associated with FSIQ for girls whose mothers had adequate MUIC_CRE_ (B = 1.31, 95%; CI: −0.41, 3.03; *p* = 0.135).

## 4. Discussion

We examined whether gestational iodine status modifies the association between prenatal fluoride exposure and preschool boys’ and girls’ intelligence in the Maternal Infant Research on Environmental Chemicals (MIREC) Study. To do so, we estimated the three-way interaction between prenatal fluoride exposure, gestational iodine status, and sex on children’s FSIQ. We found that the association between prenatal fluoride exposure and FSIQ was stronger among boys whose mothers had low urinary iodine concentrations in pregnancy compared to boys whose mothers had adequate iodine concentrations in pregnancy. These findings are consistent with previous experimental and human epidemiological studies [27,28,29,30,45] and indicate that even mildly reduced iodine levels may have biological significance when interacting with fluoride. Importantly, our findings were observed in a Canadian pregnancy sample with, on average, sufficient iodine intake (median iodine = 294 μg/g) and with 88% of women taking prenatal multi-vitamins.

Regarding potential mechanisms, experimental evidence demonstrates that prenatal exposure to both high fluoride and low iodine can induce neurochemical changes in offspring. For example, Ge et al., (2011) found that brains of rat offspring exposed to both high fluoride and low iodine in utero had different protein profiles compared with controls; proteins involved in cellular signaling and metabolism were most affected [46]. Other studies with similar experimental designs found higher levels of superoxide dismutase and malondialdehyde (biomarkers of oxidative stress), apoptosis, and histopathological changes (e.g., elongation of neural dendrites and missing nuclei) in the brains of rat offspring exposed to high fluoride and low iodine compared with those exposed to either alone [29,47,48].

The combination of low iodine and high fluoride may also adversely impact thyroid function. A prior study conducted in Canada showed that higher urinary fluoride levels in adults were associated with higher thyroid stimulating hormone levels, but only among adults who had low urinary iodine concentrations (≤0.38 µmol/L) [49]. One potential mechanism by which fluoride may interact with iodine to affect thyroid function is by inhibiting one or more enzymes involved in normal thyroid function, such as deiodinases [50]. This would increase the iodine requirement, such that the effect would be more severe in the presence of iodine deficiency. Another common hypothesis is that fluoride displaces thyroidal iodine uptake since it is more electronegative and has a lighter atomic weight than iodine [51].

Experimental studies have also shown that the effects of fluoride may be exacerbated by deficient or excess iodine [52]. For instance, Guan et al. (1988) observed decreases in T3 and T4 among adult Wistar rats with sufficient iodine intake who were exposed to fluoride at a concentration of 30 mg/L [53]. These same changes were observed among iodine-deficient rats who were exposed to fluoride at a lower concentration of only 10 mg/L. Another study with adult mice observed lower levels of triiodothyronine (T3) and thyroxine (T4) among mice with a deficient iodine intake coupled with low fluoride intake when compared with mice with a moderate iodine intake [52]. Thus, the relationship between fluoride exposure and thyroid function may differ as a function of iodine intake.

The impact of fluoride and iodine on thyroid hormones during pregnancy is especially relevant to cognitive abilities in offspring. The fetus is entirely dependent on maternal thyroid hormones until mid-gestation and continues to be partially dependent until birth [54]. Even subtle changes in maternal thyroid hormone levels in pregnancy can have adverse effects on brain structure [55,56,57] and neurodevelopment [55,56,58]. Low iodine and exposure to higher levels of fluoride in drinking water during pregnancy are both independently associated with a greater risk of developing hypothyroidism [59,60]. Our results are consistent with the combination of low iodine and high fluoride compounding thyroid disruption during fetal development, the most vulnerable period of brain development.

In our study, the interaction between fluoride and iodine was only evident in boys. This finding is consistent with a recent cross-sectional study conducted in China showing that iodine modified the susceptibility of the thyroid gland to fluoride exposure in school-aged boys, but not girls [45]. For boys with lower urinary levels of iodine, higher urinary fluoride was associated with larger thyroid volumes, whereas higher levels of iodine reduced the effects of fluoride on the thyroid. To our knowledge, no study has examined sex-specific effects of the interaction between prenatal fluoride and iodine on neurodevelopmental outcomes, but some epidemiological and animal studies of fluoride neurotoxicity found that boys are more vulnerable to prenatal fluoride exposure than girls [33]. Sex-specific effects in the interaction between fluoride and iodine, particularly among mothers with insufficient iodine intake, may disrupt in utero thyroid hormones. Given that the thyroid gland expresses estrogen and androgen receptors, boys and girls may respond differently to thyroid hormone insufficiency [61,62,63]. One study, for example, found that maternal trajectories of thyroid hormones were associated with preschool boys’ behavioural development but not girls’ [64]. Taken together, these findings indicate that future investigations of fluoride’s neurotoxicity should examine the roles of iodine intake and child sex, and whether thyroid hormones mediate the pathway for fluoride and iodine’s effects on boys’ IQ.

### Strengths and Limitations

Our study has several strengths, including a modest-sized pregnancy cohort, prospective design, state-of-the-art biomarkers of fluoride exposure and iodine status, available information on a wide array of potential maternal and child confounders, and use of standardized and valid measures of child intelligence. Our study also has limitations. Compared with the general Canadian population, women in the MIREC cohort were more educated, older, predominantly Caucasian, and more likely to be married or in common law relationships [34], which may limit the generalizability of our findings. The high use of prenatal multivitamins in our sample likely reduced the risk of moderate-to-severe iodine deficiency which may be observed in other populations. Even though we used state-of-the-art biomarkers, fluoride and iodine both have short half-lives and could therefore be impacted by recent fluoride or iodine ingestion. Further, we measured iodine and fluoride in urine spot samples instead of 24 h urine samples. We attempted to minimize this limitation by averaging urine fluoride across all three trimesters of pregnancy, and urine iodine across two trimesters of pregnancy. Nonetheless, we acknowledge that up to ten repeat spot urine samples may be needed to accurately reflect individual iodine status [65].

## 5. Conclusions

This is the first prospective epidemiological study to estimate the interplay between prenatal fluoride exposure and maternal iodine status in relation to child IQ in boys and girls. Our findings indicate that the association between prenatal fluoride exposure and full-scale intelligence previously identified in this cohort [6] was exacerbated by low maternal iodine in pregnancy among boys. These results, which were found among mother-child pairs living in fluoridated and non-fluoridated communities in Canada, underscore the importance of sufficient iodine intake in pregnancy to minimize the neurotoxicity of fluoride in boys.

## Figures and Tables

**Figure 1 nutrients-14-02920-f001:**
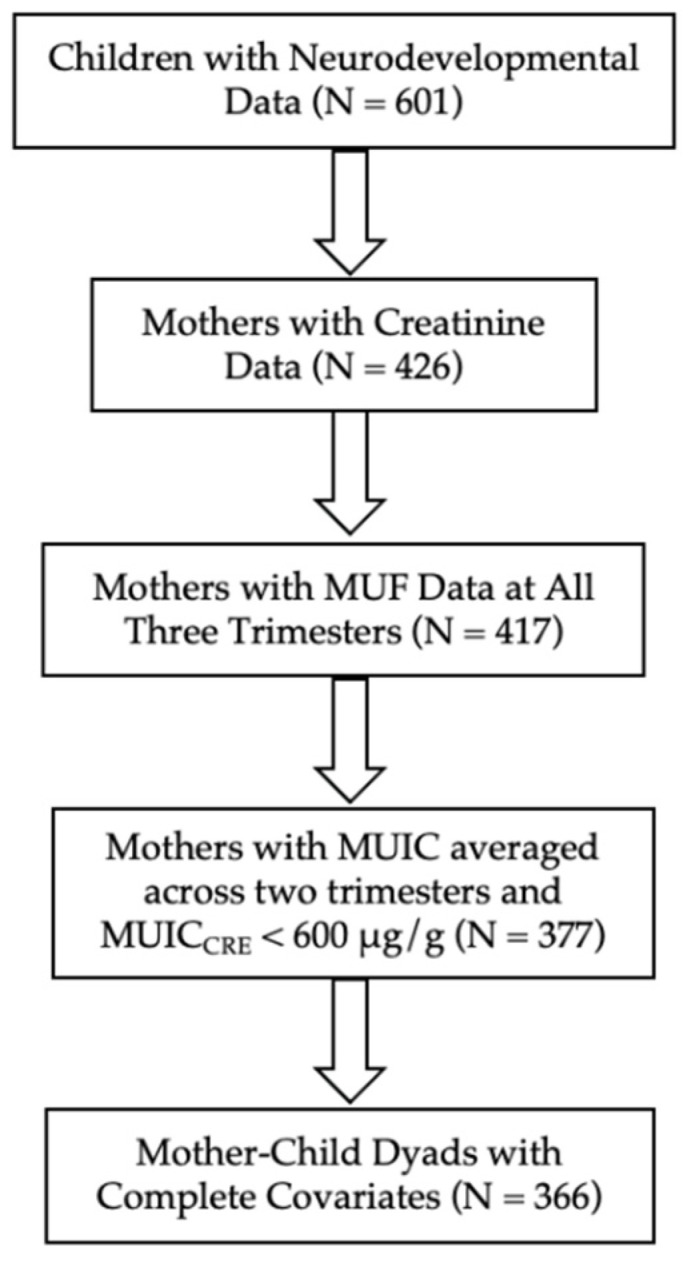
Sample flow chart.

**Figure 2 nutrients-14-02920-f002:**
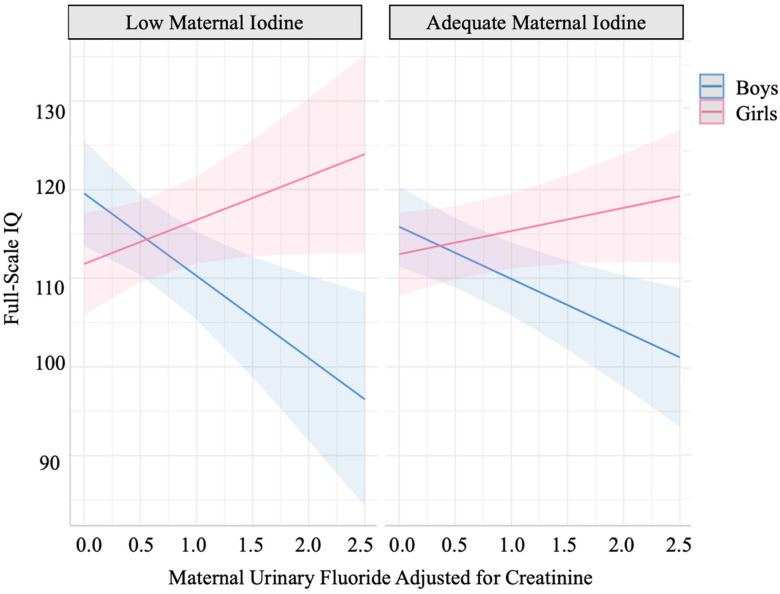
Model—implied three-way interaction between maternal urinary fluoride (MUF_CRE_), maternal urinary iodine concentration (MUIC_CRE_) and child sex. Every 0.5 mg/g increase in MUF_CRE_ was significantly associated with a 4.65- and −2.95-point lower FSIQ score for boys whose mothers had low MUIC_CRE_ or adequate MUIC_CRE_, respectively. MUF_CRE_ was marginally associated with FSIQ for girls whose mothers had low MUIC_CRE_ and not significantly associated with FSIQ for girls whose mothers had adequate MUIC_CRE_.

**Table 1 nutrients-14-02920-t001:** Demographic Characteristics of those with Complete Data (*N* = 366) and Incomplete Data (*N* = 211).

Demographic Characteristic (Mean ± SD or *N* (%))	Complete Data (*N* = 366)	Incomplete Data (*N* = 211)	*p*
Mothers			
Maternal Age (years)	32.50 ± 4.51	32.55 ± 4.62	0.899
Married or Common Law	353 (96.54)	205 (97.16)	0.646
White	334 (91.26)	181 (85.78)	0.041
Bachelor’s Degree or Higher	243 (66.39)	142 (67.30)	0.824
Taking a prenatal multivitamin	319 (87.40)	175 (82.94)	0.140
HOME Score	47.23 ± 4.44	47.40 ± 4.10	0.649
Children			
Male	186 (50.82)	98 (46.44)	0.311
Age at Testing (years)	3.44 ± 0.32	3.40 ± 0.31	0.144

Abbreviations: HOME = Home Observation Measurement of the environment.

**Table 2 nutrients-14-02920-t002:** MUF_CRE_, MUIC_CRE_, and Full-Scale IQ by sex.

Urinary Measurement	All		Boys		Girls		
	*n*	Median (IQR)	*n*	Median (IQR)	*n*	Median (IQR)	*p* ^1^
MUF_CRE_ (mg/g)	366	0.61 (0.49)	186	0.63 (0.52)	180	0.61 (0.48)	0.538
MUIC_CRE_ (μg/g)	366	294 (181)	186	309 (181)	180	287 (203)	0.059
Low	86	148 (47)	31	131 (73)	55	152 (37)	0.083
Adequate	280	341 (165)	155	348 (187)	125	336 (146)	0.893
**Outcome**	** *n* **	**Mean ± SD**	** *n* **	**Mean ± SD**	** *n* **	**Mean ± SD**	***p* ^1^**
FSIQ	366	107.46 ± 13.75	186	105.25 ± 14.90	180	109.75 ± 12.09	0.002

Low MUIC_CRE_ < 200 μg/g, Adequate MUIC_CRE_ ≥ 200 & < 600 μg/g; Abbreviations: MUF_CRE_ = Maternal urinary fluoride corrected for creatinine; MUIC_CRE_ = maternal urinary iodine concentration corrected for creatinine; FSIQ = Full-Scale IQ. ^1^ Comparing boys with girls.

**Table 3 nutrients-14-02920-t003:** Results of the three-way interaction model.

Variable	B	SE(B)	*p*
MUF_CRE_ (mg/g)	−5.89	1.85	0.002
MUIC_CRE_ (μg/g)	−0.03	0.01	0.023
Sex	−3.09	2.17	0.155
MUF_CRE_ × MUIC_CRE_	0.02	0.01	0.042
MUF_CRE_ × Sex	8.51	2.40	<0.001
MUIC_CRE_ × Sex	0.03	0.02	0.042
MUF_CRE_ × MUIC_CRE_ × Sex	−0.04	0.02	0.019

Note. SE: Standard Error, *R*^2^ = 0.28, *F* (15, 350) = 8.97, *p* < 0.001; Abbreviations: MUF_CRE_ = maternal urinary fluoride corrected for creatinine; MUIC_CRE_ = maternal urinary iodine concentration corrected for creatinine. Model adjusted for maternal level of education, maternal ethnicity, HOME score, and study site. MUIC_CRE_ is centered around the “adequate” level of iodine, and boys are coded as the reference. The coefficient for MUF_CRE_ represents the association between MUF_CRE_ and FSIQ for a boy whose mother had an adequate level of MUIC_CRE_ during pregnancy.

## Data Availability

Anonymized data described in the manuscript are available to qualified investigators with IRB approval.

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
