# Peer review of "Iodine Status Modifies the Association between Fluoride Exposure in Pregnancy and Preschool Boys’ Intelligence"

_nutrients, 2022, doi:10.3390/nu14142920_

Round 1

Reviewer 1 Report

Drinking water was first fluorinated in the small town of Grand Rapids in Michigan, USA, in January 1945. Since then, the results of hundreds of clinical studies have been published, which have shown with certainty that this method is justified. An analysis of the results obtained from 113 clinical studies conducted in 23 countries leads to the conclusion that fluoridation of drinking water reduces caries on both deciduous (40-49%) and permanent teeth (50-59%). However, research in the meantime has shown that higher levels of fluoride exposure during pregnancy were associated with lower IQ scores in children measured at age 3 to 4 years.  Now the authors of the manuscript: "Iodine Status Modifies the Association Between Fluoride Exposure in Pregnancy and Preschool Boys’ Intelligence" goes further and find a link between higher amounts of fluoride and lower amounts of iodine with preschool boys’ intelligence. Through a series of researches on this topic, the authors open a significant topic of nutritional intervention for the sake of one goal, without thinking about further consequences. That is why I sincerely support the publication of this paper, which was also written and discussed correctly.

Please just improve quality of Figures!

Author Response

We would like to start off by thanking both reviewers for taking the time to review our manuscript and for providing us with helpful feedback and commentary. We have addressed their questions and concerns below. 

Reviewer 1:

Reviewer’s comment: Please revise the quality of the figures

Author’s reply: We have included higher quality figures in the manuscript.

Reviewer 2 Report

The authors present the results of a very interesting and original clinical study attempting to catch a complex relationship between maternal iodine status, maternal fluoride exposure in pregnancy and the intelligence of preschool children as assessed by FSIQ scale. Although the relevant associations were not apparent for girls, for boys whose mothers had low iodine, an increase in the maternal urine fluorine was associated with a decrease in FSIQ score. The study is distinguished by a thoughtful design (including careful selection of the participants). There are several points addressing which can further improve the presentation of the paper.

1. The most important question is an apparent lack of association for girls. Actually, the results of analysis of association between iodine and FSIQ for girls were on the verge of statistical significance (B = 2.48; 95% CI: -0.31, 5.26; p = 0.081) which certainly merits the terms like "trend in association", "tentative association" and not just "not significant" (just read what Sir Ronald Fisher wrote on the arbitrary 'statistical significance' of p=0.05...). Although the authors did a good work on exploratory data analysis, there is more can be done to elucidate the issue. For example, would if the girls from mothers with iodine >600 units be included in the study? What other covariates might influence the association for girls (may be some details of anamnesis)? What would be the results of perturbation analysis, that assess "stability" of statistical associations (as described in "Sensing the change from molecular genetics to personalized medicine", Nova Science, 2009, ISBN-10: 1-60741-704-9. See also Modern Pharmacoeconomics and Pharmacoepidemiology 2019; 12 (2): 91-114. DOI: 10.17749/2070-4909.2019.12.2.91-114)?
2. The section on mechanisms of the interaction between the iodine and fluorine has to be expanded. What could be the molecular mechanisms involved a direct iodine displacement by fluoride which is the most electronegative element, or an increased flow of fluoride into cells through the iodine channels when there is lack of iodine ion in the bodily fluids (increased inflow of fluorine then causes neuronal damage on cellular level), or some proteomic or transcriptomic effects of either/both iodine and fluoride etc.
3. The abbreviations are not systematic: why "maternal urinary iodine concentration" is abbreviated as "UIC" while maternal urinary fluoride as "MUF"? Wouldn't "MUIC" and "MUF" be better?
4. "Two of the spot urine samples (0.002%)" - here is an obvious error since if 2 patients correspond to 0.002%, then 50000 patients were screened ? Perhaps, "0.2%"?

Author Response

We would like to start off by thanking both reviewers for taking the time to review our manuscript and for providing us with helpful feedback and commentary. We have addressed their questions and concerns below:

Reviewer 2:

Reviewer’s comment: The most important question is an apparent lack of association for girls. Actually, the results of analysis of association between iodine and FSIQ for girls were on the verge of statistical significance (B = 2.48; 95% CI: -0.31, 5.26; p = 0.081) which certainly merits the terms like "trend in association", "tentative association" and not just "not significant" (just read what Sir Ronald Fisher wrote on the arbitrary 'statistical significance' of p=0.05...). Although the authors did a good work on exploratory data analysis, there is more can be done to elucidate the issue. For example, would if the girls from mothers with iodine >600 units be included in the study? What other covariates might influence the association for girls (may be some details of anamnesis)? What would be the results of perturbation analysis, that assess "stability" of statistical associations (as described in "Sensing the change from molecular genetics to personalized medicine", Nova Science, 2009, ISBN-10: 1-60741-704-9. See also Modern Pharmacoeconomics and Pharmacoepidemiology 2019; 12 (2): 91-114. DOI: 10.17749/2070-4909.2019.12.2.91-114)?

Author’s Reply: We have revised the wording in the abstract and results to suggest a marginally significant association in girls with low maternal urinary iodine. Although we respect your recommendation to conduct exploratory data analysis to elucidate this issue, we have chosen to not go further to try to explain why we found a trend in girls, as we believe this marginal association does not warrant an over-interpretation of the findings. Further, we have chosen not to include mothers with iodine >600, given that too much iodine can also be detrimental to children’s intelligence (Li et al., 2022), so it would not make biological sense to study them in this type of analysis. In terms of using a perturbation analysis, to our understanding, this is used to approximate solutions of nonlinear equations when exact solutions can’t be obtained. We do not believe this type of analysis would be appropriate given that we are using a linear regression model and the assumption of linearity was met.

Li, F.; Wan, S.; Zhang, L.; Li, B.; He, Y.; Shen, H.; Liu, L. A Meta-Analysis of the Effect of Iodine Excess on the Intellectual Development of Children in Areas with High Iodine Levels in Their Drinking Water. Biol. Trace Elem. Res. 2022, 200, 1580–1590, doi:10.1007/s12011-021-02801-3.

Reviewer’s comment: The section on mechanisms of the interaction between the iodine and fluorine has to be expanded. What could be the molecular mechanisms involved a direct iodine displacement by fluoride which is the most electronegative element, or an increased flow of fluoride into cells through the iodine channels when there is lack of iodine ion in the bodily fluids (increased inflow of fluorine then causes neuronal damage on cellular level), or some proteomic or transcriptomic effects of either/both iodine and fluoride etc.

Author’s Reply: Thank you very much for your comment. We added a paragraph commenting on potential mechanisms by which fluoride and iodine may interact.

Reviewer’s comment: The abbreviations are not systematic: why "maternal urinary iodine concentration" is abbreviated as "UIC" while maternal urinary fluoride as "MUF"? Wouldn't "MUIC" and "MUF" be better?

Author’s Reply: The abbreviation for UIC has been revised to MUIC.

Reviewer’s comment: "Two of the spot urine samples (0.002%)" - here is an obvious error since if 2 patients correspond to 0.002%, then 50000 patients were screened ? Perhaps, "0.2%"?

Author’s Reply: Thank you. We have corrected the value. It should be 0.04% as there were 2 out of 5283 samples that surpassed the highest concentration standard of the instrument.

Editor Comments:

Editorial comment: Thank you for the excellent submission.   Not sure how surprising this is - but I found internal documents from the 1940s on how mandating iodine was the solution for the effect of fluoride on the thyroid (no mention of intelligence).  

Author’s comment: Thank you very much for sharing this information with us. 

Round 2

Reviewer 2 Report

Presentation of the paper improved, the paper can be published after minor technical editing.